# The Prediction of the Potentially Suitable Distribution Area of *Cinnamomum mairei* H. Lév in China Based on the MaxEnt Model

Shuai Qi [1], Wei Luo [1], Ke-Lin Chen [1], Xin Li [1], Huo-Lin Luo [2,*], Zai-Qiang Yang [3,*] and Dong-Mei Yin [1,*]

[1] Shanghai Institute of Technology, College of Ecological Engineering and Technology, Shanghai 201418, China; don_shuai@163.com (S.Q.); weichanyi183@163.com (W.L.); chenkelin@126.com (K.-L.C.); 15543358779@163.com (X.L.)

[2] School of Life Sciences, Nanchang University, Nanchang 330031, China

[3] Collaborative Innovation Center on Forecast and Evaluation of Meteorological Disasters, Nanjing University of Information Science and Technology, Nanjing 210044, China

* Correspondence: 572460991@163.com (H.-L.L.); yzq@nuist.edu.cn (Z.-Q.Y.); yindm@sit.edu.cn (D.-M.Y.); Tel.: +86-21-58731129 (Z.-Q.Y.); +86-21-60873096 (D.-M.Y.); Fax: +86-21-60873128 (D.-M.Y.)

**Abstract:** *Cinnamomum mairei* H. Lév is a rare and valuable medicinal and timber species in China. It not only has a narrow distribution, but also has few resources, is an endangered species, and is a nationally protected plant. Climate change impacts the growth and development of plants; therefore, it is of great practical significance to predict the current and future distribution of *C. mairei* H. Lév in suitable areas of China and to protect these endangered plants. In this study, the MaxEnt model was used to predict the suitable growing areas for *C. mairei* H. Lév according to six environmental factors (the temperature seasonality, max. temperature in the warmest month, min. temperature in the coldest month, precipitation seasonality, precipitation in the coldest quarter and aspect), and three different climate models (SSP126, SSP245, and SSP585) were simulated for three periods (the 2030s, 2050s, and 2070s). In the present study, the suitable ecological environment for *C. mairei* H. Lév comprised the following: a min. temperature in the coldest month from −0.63 to 4.36 °C, temperature seasonality from 130.67 to 642.58, a max. temperature in the warmest month from 28.86 to 45.97 °C, and precipitation in the coldest quarter from 40.12 to 101.13 mm. Highly suitable habitats for *C. mairei* H. Lév are located in the Yunnan Province, Guizhou Province, Sichuan Province, and Chongqing City, China (southwestern part of China), and to a lesser extent in the Xizang Province and Shaanxi Province, China. The moderately suitable habitat district overlaps with the highly suitable habitat district, and a small number of suitable habitats are distributed in Guangxi Province, Hunan Province, Hubei Province, and Henan Province. In the future, the highly suitable areas for *C. mairei* H. Lév will increase slightly, and the gravity points will shift toward northeast China. Our simulations are helpful for understanding the geoecological characteristics of this species and provide a basis for regional projections of this species under current and future climate change scenarios in China. It is proposed to establish nature reserves for *C. mairei* H. Lév in the Chongqing, Yunnan, Sichuan and Guizhou provinces of China.

**Keywords:** climate change; *Cinnamomum mairei* H. Lév; potential geographical distribution; suitable habitat; MaxEnt





## 1. Introduction

Many factors affect the distribution of plant species, including adaptations to the ecological environment and the interactions between living and abiotic species [1]. Climate change affects the living area and distribution of species, requiring species to adapt to environmental changes [2]. In recent decades, climate change has had a significant impact on the entire ecosystem, including changes in the population structures and life history,

geographical ranges, and the ecosystem structure and function, which are closely related to a global population decline and species extinction [3]. In a fragile ecological environment, plant populations are unable to adapt to rapid climate change. As a result, in the face of decoupling climate change and local climate adaptation, the adaptability of plants and the viability of whole species are greatly reduced, and the internal genetic composition of species is changed [4]. Climate change may alter the vegetation structure and damage ecosystems, resulting in habitat fragmentation and loss as well as a reduced habitat distribution [5,6]. Understanding the relationship between a species or community and its environment is a key concept in ecology and conservation, while the use of species distribution models (SDMs) for this purpose has advanced the understanding of many ecological issues and is a key tool in predicting how species respond to environmental change [7].

Currently, SDMs have become a common tool in spatial ecology to predict species distributions and environmental habitats [8–10]. Species distribution models can predict the spatial and temporal distribution of a species by correlating information about known species with environmental variables that affect their growth [11]. GARP [12], MaxEnt [13], Domain [14], and Bioclim [15], for example, are common species distribution models. MaxEnt benefits from its ability to model presence-only (PO) data [16] and is thought to be robust to small sample sizes [17] as well as able to model the complex, non-linear relationships between the response variables and predictors [9]; however, it is the ease and simplicity of its implementation that has propelled MaxEnt to be the most prominent, widely-used SDM technique in scientific research [18].

*C. mairei* H. Lév, also known as Yinyegui (in Chinese), which can grow up to 16 m in height, is a precious medicinal and timber tree in China. *C. mairei* H. Lév was most recently assessed for inclusion on the International Union for Conservation of Nature (IUCN) Red List of Threatened Species in 1998 (https://www.iucn.org/, accessed on 11 June 2022). *C. mairei* H. Lév is listed as Endangered under criteria B1 + 2c. It not only has a narrow distribution, but also has few resources, is an endangered species, and is a protected plant in China. We investigated the distribution sites of *C. mairei* H. Lév using three methods: website data, field surveys, and literature collected from the Web of Science. It is mainly concentrated in the Chinese region, therefore, China was chosen as our study area. Within China, it is distributed in northeast Yunnan and western Sichuan. Its dry bark is medicinal, and is able to disperse cold and relieve pain [19]. At present, scholars mainly study the chemical composition and medicinal value of *C. mairei* H. Lév [19], and there is almost no research on the distribution patterns of this species. In the context of rapid climate and environmental change, it is particularly urgent to study the migration and dispersal of the species and its response and adaptation to climate warming [20]. The response of this species to climate change has not been studied in the global or Chinese literature, and the potential range of this species is unknown. Usually, the distribution of plants has a very important connection with climate [21]; therefore, this problem can be solved using the MaxEnt model, which simulates the potential global distribution under the current and future climatic conditions using the distribution points and environmental variables of *C. mairei* H. Lév. Additionally, the main climatic factors of the current potential distribution pattern were identified.

Therefore, it is of great significance to predict the distribution of potentially suitable areas for *C. mairei* H. Lév under the conditions of global climate change. This study adopted the MaxEnt model and ArcGIS technology for parameter optimization, climate variables, and topography to research the potential current and future growing regions of *C. mairei* H. Lév in the next 60 years in China, which will help to further the biological lay research on *C. mairei* as well as provide effective species protection. Additionally, sustainable ecological development is very important and the objectives of this study were as follows: (1) to simulate the effects of climatic factors and topographic factors on the distribution pattern of *C. mairei* H. Lév to obtain the current spatial and temporal distribution information for *C. mairei* H. Lév; (2) to determine the important environmental variables related to the potential distribution of *C. mairei* H. Lév; and (3) to forecast the potential distribution of

*C. mairei* H. Lév in China in the next 60 years. The results of this study will help us to better understand the adaptation and change process and mechanism of *C. mairei* H. Lév under complex climate and environmental conditions and provide scientific support for the conservation and research of *C. mairei* H. Lév in China.

## 2. Materials and Methods

### 2.1. Occurrence Data

The Chinese *C. mairei* H. Lév data were mainly obtained from a literature search, the China Nature Reserve Resources Platform (http://www.papc.cn/, accessed on 20 December 2021), the Chinese Virtual Herbarium (CVH) (https://www.cvh.ac.cn/, accessed on 20 December 2021), and the Global Biodiversity Information Facility (GBIF) (https://www.gbif.org/ accessed on 20 December 2021) as well as field surveys. To remove spatial autocorrelation, a threshold value of 10 km was set to delete the data. A total of 22 valid data points were retained, and the coordinate system was unified (Figure 1).

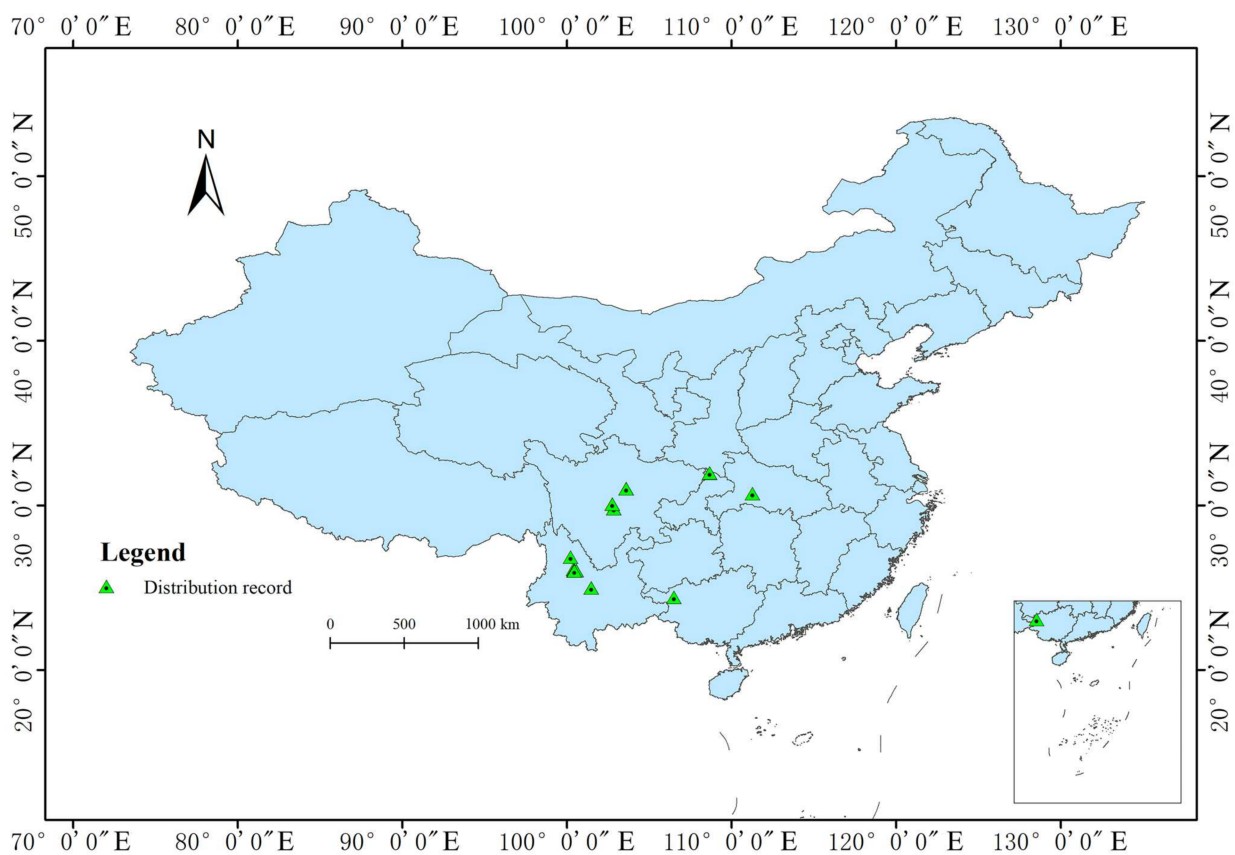

**Figure 1.** Distribution of *C. mairei* H. Lév data points. This map was made based on the standard map No. GS (2019) 1822 downloaded from the standard map service website of the National Administration of Surveying, Mapping and Geographic Information. The base map is unchanged, and the geographical coordinate is WGS84.

### 2.2. Environmental Variables and Processing

Climate data from the WorldClim website (https://www.worldclim.org/, accessed on 20 December 2021) containing 19 climate variables and geographic data from the database of the Chinese Academy of Sciences with a resolution of 30 m by 30 m were analyzed using spatial analysis tools, and DEM data were used for the slope extraction, slope direction, and to determine the elevation data. A total of 22 pieces of environmental data were obtained to predict the geographical distribution of *C. mairei* H. Lév. ArcGIS10.6 was used to convert all of the environmental data into the ASCII format for the MaxEnt model. Excessive data

may lead to over-fitting after operation in an imported model; therefore, environmental variables with a high autocorrelation were needed to be excluded (Figure 2). A Pearson correlation analysis was used to exclude the variables with $|r| \geq 0.8$ in SPSS (Version number: IBM SPSS Statistics 26; Creator: IBM Corporation; Location: city of Armonk, USA), and six environmental variables were retained for the model analysis. These six environmental variables were bio-4 (temperature seasonality), bio-5 (max. temperature in the warmest month), bio-6 (min. temperature in the coldest month), bio-15 (precipitation seasonality), bio-19 (precipitation in the coldest quarter), and aspect, respectively.

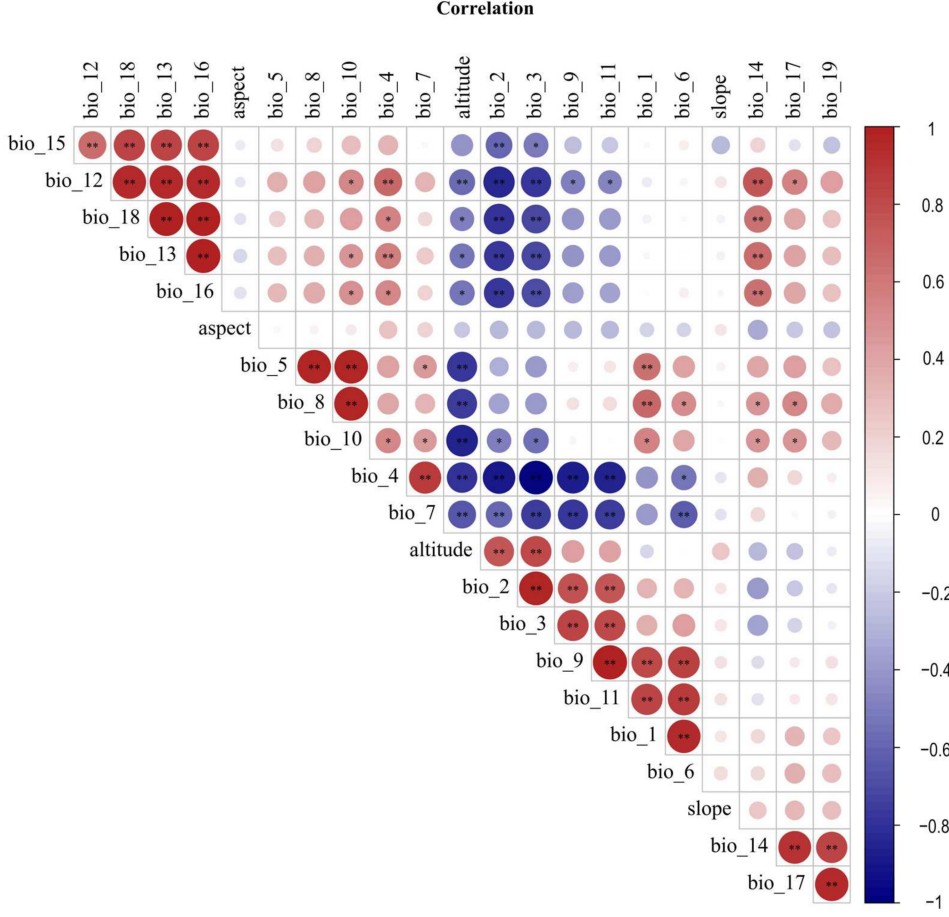

**Figure 2.** Correlation analysis of environmental variables. Pearson's correlation coefficient, which measures the degree of the linear correlation between two quantitative variables, was used for the correlation analysis. Red indicates a positive correlation and blue indicates a negative correlation; * significant correlation at the 0.05 level; ** significant correlation at the 0.01 level.

### 2.3. MaxEnt Model Accuracy Verification

The *C. mairei* H. Lév data points and six environment variables with self-correlation were input into the MaxEnt 3.4.1 model, and the model was set so that 75% of the distribution points were used to train the data set to establish the model, and the remaining 25% of the distribution points were used as the test data set to verify the model. The maximum number of iterations was $1 \times 106$, and the response curve, the ROC curve, and jackknife test were set, while the remainder were set to the default setting. To ensure the stability of the model, 10 iterations were carried out. The area under the curve (AUC) of the receiver operating characteristic was used to evaluate the quality of the model results [22], and the evaluation criteria are shown in Table 1.

**Table 1.** AUC evaluation criteria table.

| AUC Value | Evaluation Criterion |
|---|---|
| 0.5~0.7 | Poor performance |
| 0.7~0.9 | Satisfactory performance |
| 0.9~1.0 | High performance |

### 2.4. Statistical Analysis and Suitable Habitat Grade Classification

The model's ASCII files were input into the ArcGIS software (Version number: ArcGIS 10.6; Creator: Esri Corporation; Location: Redlands, USA) and turned into raster layers, overlaid with the map of China, and the raster layers were then reclassified. According to the classification method of IPCC for assessing the possibility, it was divided into four categories [23]: an unsuitable zone ($p < 0.05$), low fitness zone ($0.05 \leq p < 0.33$), moderate fitness zone ($0.33 \leq p < 0.66$), and high fitness zone ($p \geq 0.66$), and the areas of each zone were calculated.

### 2.5. Future Climate

The new Intergovernmental Panel on Climate Change (IPCC) has proposed shared socioeconomic pathways (SSPs) to project climate change scenarios [24] and the SSPs are designed based on a global development framework [25]. They have been widely used to project social and climate scenarios on a global scale [26] and have achieved good results for comprehending future uncertainties. The SSPs include the four emissions scenarios [27]: (1) radiative forcing stabilizes at 8.5 W m$^2$ in 2100 (SSP5-8.5); (2) radiative forcing stabilizes at 7.0 W m$^2$ in 2100 (SSP3-7.0); (3) radiative forcing stabilizes at 4.5 W m$^2$ in 2100 (SSP2-4.5); and (4) radiative forcing stabilizes at 2.6 W m$^2$ in 2100 (SSP1-2.6). Compared to the analysis tools used to predict future climate scenarios, the SSPs have some advantages: (1) SSPs can comprehensively reflect the complexity of regional development, as their paths take into account various factors of local circumstances, such as population, economy, policy, technology, the environment, and resources [28]. (2) SSPs in regional impact assessment models usually have a high resolution and are corrected by historical data. (3) SSPs have better precision than global models. Therefore, to determine the future distribution of a species under different climate trajectories, the habitat suitability distribution of *C. mairei* H. Lév in the 2030s (2021–2040), 2050s (2041–2060) and 2070s (2061–2080) was simulated with SSP1-2.6, SSP2-4.5, and SSP5-8.5.

## 3. Results

### 3.1. Model Performance

The model that was generated was evaluated by calculating the AUC (area under the curve) of the plot of the receiver operating characteristic (ROC) [29]. The ROC curve results for 10 simulations (Figure 3) showed that the AUC of the training set and the AUC of the test set of the prediction model were 0.971 and 0.929, respectively, and the mean value of the AUC was 0.955. Both values being greater than 0.9 indicated that the model was extremely accurate and had a high reference value, meaning that the model could be used to predict the potential distribution in China.

### 3.2. Environmental Variable Importance

According to the MaxEnt model, the leading environmental variables are the min. temperature in the coldest month, the temperature seasonality, and the precipitation in the coldest quarter (Table 2), which make a cumulative contribution of more than 96.9%, contributing 70.6%, 15.5%, and 12.4%, respectively. The results indicate that temperature was the primary factor affecting the distribution of *C. mairei* H. Lév followed by rainfall, and that the other environmental factors had little impact on the distribution. The maximum temperature in the hottest month contributed 2%, the seasonal variation in rainfall contributed 0.6%, and the aspect contributed 0.5%.

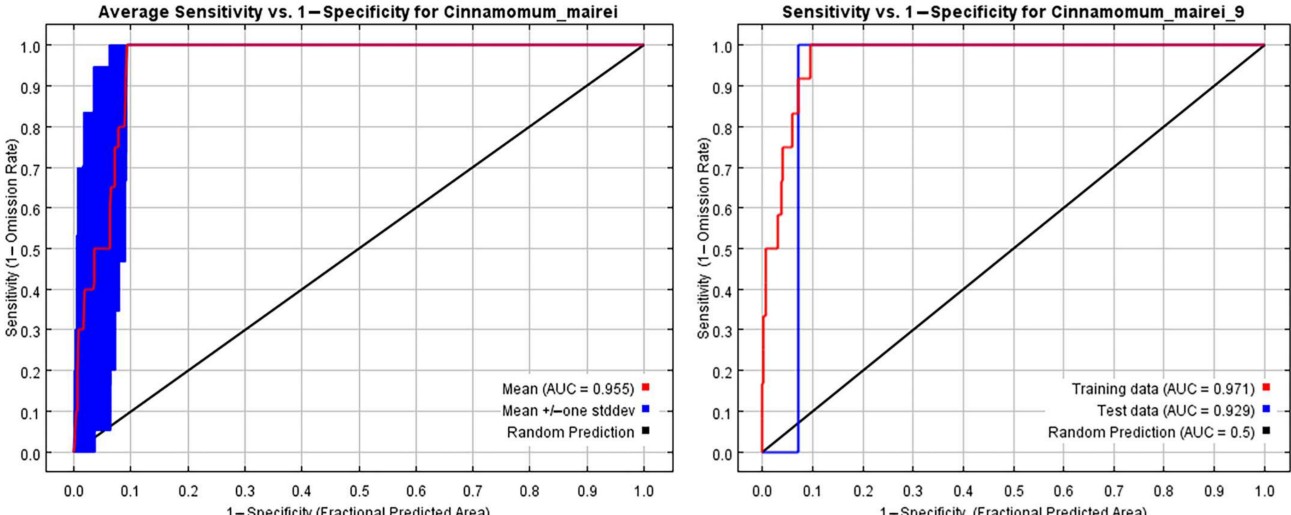

**Figure 3.** ROC curve of 10 simulations. A receiver operating characteristic curve (ROC curve) is a graphical plot that illustrates the diagnostic ability of a binary classifier system as its discrimination threshold is varied.

**Table 2.** Contribution rate of dominant environmental variables.

| Variable | Percent Contribution/% | Permutation Importance/% |
|---|---|---|
| bio_6 | 69 | 70.6 |
| bio_4 | 15.5 | 13.2 |
| bio_19 | 12.4 | 13.4 |
| bio_5 | 2 | 2.2 |
| bio_15 | 0.6 | 0.3 |
| aspect | 0.5 | 0.3 |

According to the jackknife test (Figure 4), the environmental variables that play a major role are the min. temperature in the coldest month (bio_6), temperature seasonality (bio_4), the precipitation of the coldest quarter (bio_19), precipitation seasonality (bio_15), the max. temperature in the warmest month (bio_5), and the aspect. The most important environmental factors are the min. temperature in the coldest month, temperature seasonality, and the precipitation of the coldest quarter.

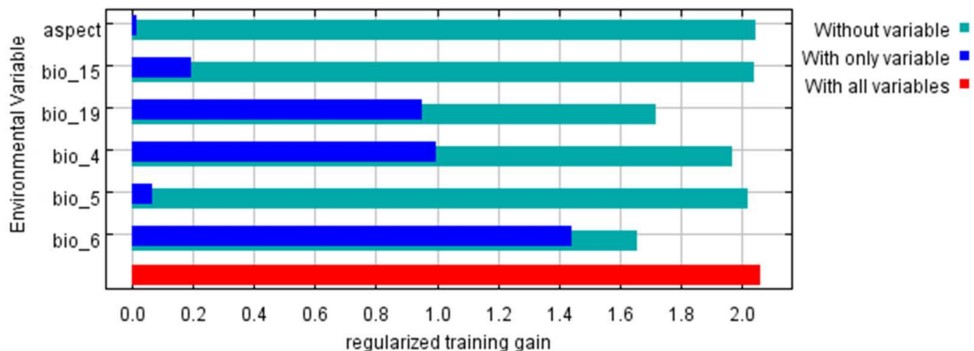

**Figure 4.** Jackknife test of environmental variables affecting the potential distribution of *C. mairei* H. Lév.

According to the single-factor contribution rate of the MaxEnt model (Figure 5), the environmental variables for the optimal growing area of *C. mairei* H. Lév are a min. temperature in the coldest month (bio_6) of −0.63~4.36 °C; temperature seasonality (bio_4)

of 130.67~642.58; a max. temperature in the warmest month (bio_5) of 28.86~45.97 °C; and a precipitation in the coldest quarter (bio_19) of 40.12~101.13 mm. The distribution probability of *C. mairei* H. Lév in this ecological environment is $p \geq 0.66$.

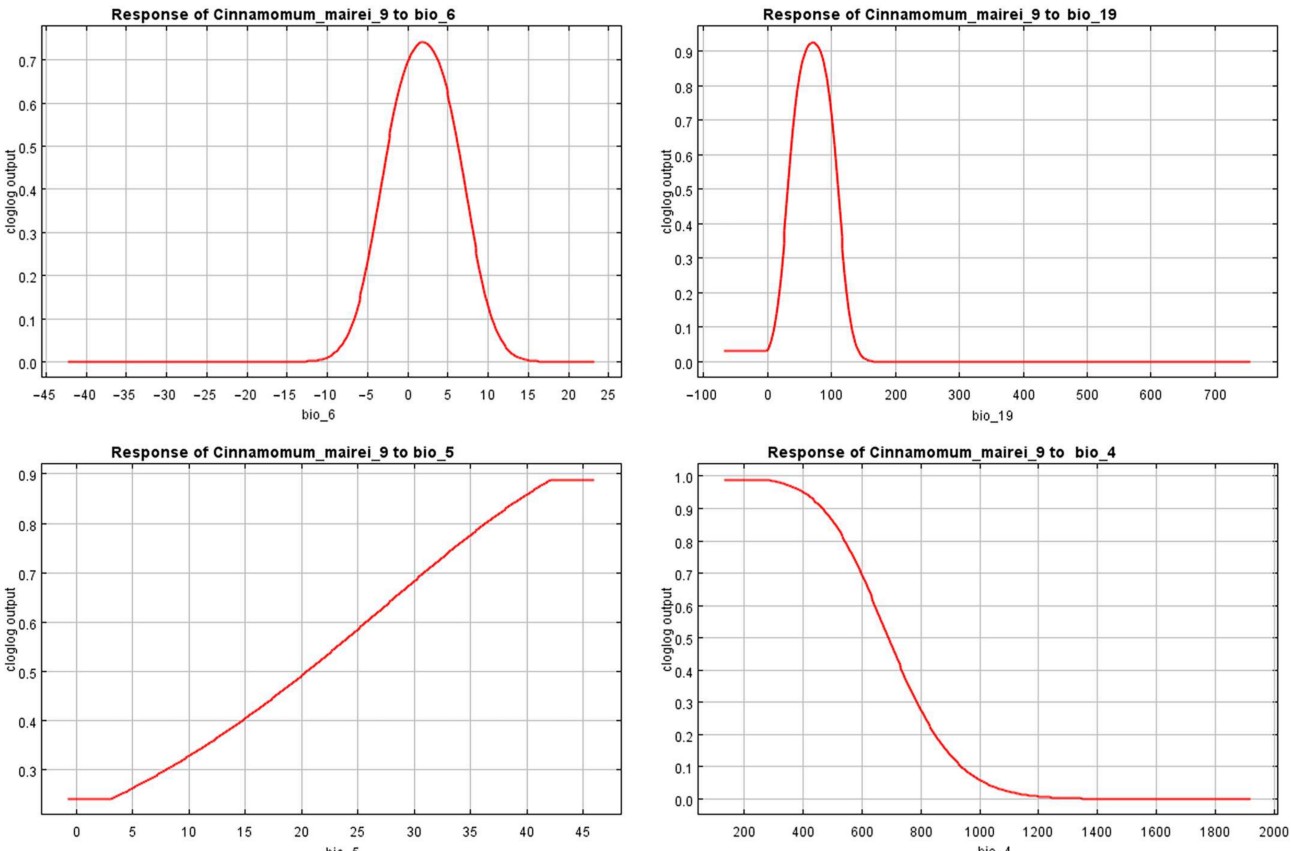

**Figure 5.** Response curve of dominant environment variables.

### 3.3. Prediction of the Potential Geographic Distribution of C. mairei H. Lév under Current Climatic Conditions

It can be seen from Figure 6 that *C. mairei* H. Lév is generally distributed in southwest China, mainly in the whole Yunnan Province, northwest and southwest Guizhou, and central Chongqing. Southern and eastern Sichuan, southeastern Xizang, a small portion of northwestern Hubei, and southern Shaanxi are also suitable locations. The most suitable area for *C. mairei* distribution has an area of $39.32 \times 10^4$ km$^2$, the moderately suitable area has an area of $53.75 \times 10^4$ km$^2$, and the least suitable area has an area of $77.76 \times 10^4$ km$^2$.

### 3.4. Potential Habitat Changes of C. mairei H. Lév under Future Climate Scenarios

Using the environmental data from 2021–2040, 2041–2060, and 2061–2080, the future distribution pattern of *C. mairei* H. Lév was predicted, and three climatic conditions: SSP126, SSP245, and SSP585, were used to simulate those time periods, respectively. The potential distribution of China's *C. mairei* H. Lév from 2021 to 2080 was predicted to be as shown in Figure 7.

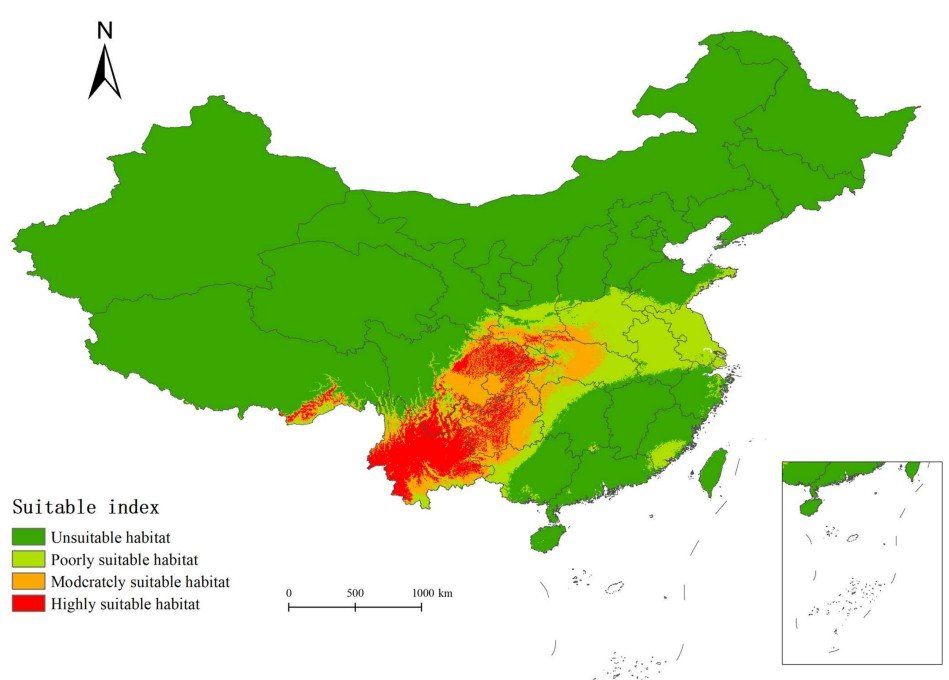

**Figure 6.** Distribution of ecologically suitable areas for *C. mairei* H. Lév in China. This map was made based on the standard map No. GS (2019)1822, which was downloaded from the standard map service website of the National Administration of Surveying, Mapping and Geographic Information. The base map is unchanged, and the geographical coordinate is WGS84.

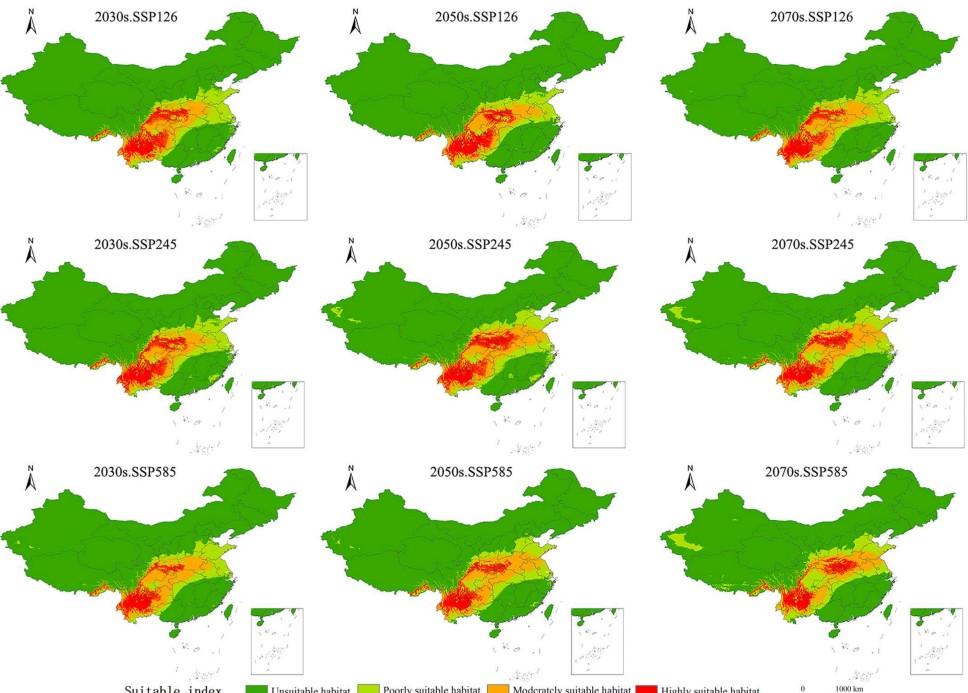

**Figure 7.** Prediction of suitable areas for *C. mairei* H. Lév in China in the future. This map was made based on standard map No. GS (2019) 1822, which was downloaded from the standard map service website of the National Administration of Surveying, Mapping and Geographic Information. The base map is unchanged, and the geographical coordinate is WGS84.

In Table 3, the suitable area in China for *C. mairei* H. Lév changed and was gradually moved north and east. Under the different climate conditions, the highly suitable area

for *C. mairei* H. Lév varies greatly within China. Under the SSP126 scenarios, the area of the highly suitable area will continue to expand, increasing by 11.63% and reducing by 0.38% compared to the area in the 2070s. Under the SSP245 scenarios, the areas that are the most suitable for *C. mairei* H. Lév growth in the 2030s, 2050s, and 2070s increased by 15.8%, 27.86%, and 13.34%, respectively; under the SSP585 scenarios, the growth of S. *C. mairei* H. Lév was restricted, and the most suitable area decreased, with a maximum reduction of 13.15% being observed in the 2030s.

**Table 3.** Size of suitable areas in different periods and scenarios.

| Period | Highly Suitable Habitat/$10^4$ km$^2$ | Variation Tendency/% |
|---|---|---|
| Current | 39.32 | 0% |
| 2030s-126 | 43.48 | 10.58% |
| 2030s-245 | 45.53 | 15.80% |
| 2030s-585 | 34.15 | −13.15% |
| 2050s-126 | 43.89 | 11.63% |
| 2050s-245 | 50.27 | 27.86% |
| 2050s-585 | 36.91 | −6.13% |
| 2070s-126 | 39.17 | −0.38% |
| 2070s-245 | 44.57 | 13.34% |
| 2070s-585 | 37.24 | −5.30% |

*3.5. The Changing Trend of Highly Suitable Habitat Gravity Points of C. mairei H. Lév under Different Climatic Conditions*

The extraction tool in the ArcGIS software was used to determine the highly suitable areas and calculate the center of gravity points in each period (Figure 8). The center of mass in the high suitability area under the current climate conditions is 105.047° E, 28.768° N. Under the climate conditions of SSP126, the centers of the high suitability zones in the 2030s, 2050s, and 2070s are 105.540° E, 29.140° N; 105.552° E, 29.237° N; and 105.345° E, 28.879° N, respectively. Compared to the current center of mass, the center of mass shifted to the northeast by 63.39 km, 71.63 km, and 31.63 km, respectively. Under the climate conditions of SSP245, in the 2030s, 2050s, and 2070s, the centers of the high suitability zones are 105.810° E, 29.352° N; 106.710° E, 29.658° N; and 105.947° E, 29.502° N, respectively. Compared to the current northeast direction, the centroid is offset by 98.62 km, 189.47 km, and 119.59 km, respectively; under the climate conditions of SSP585, the centers of the high suitability zones in the 2030s, 2050s, and 2070s are 104.324° E, 28.856° N; 105.020° E, 29.124° N; and 105.041° E, 30.585° N, respectively. The center of mass shifted 71.23 km, 39.51 km, and 201.43 km from the current north direction.

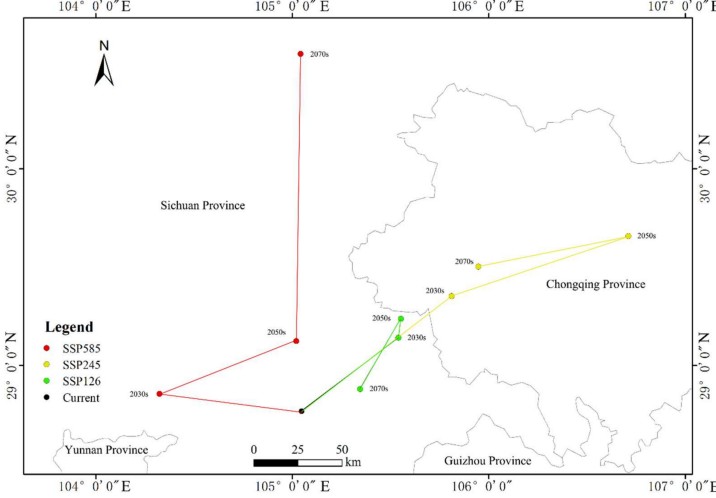

**Figure 8.** The change trends in the gravity points of the most suitable areas for *C. mairei* H. Lév under different climatic conditions.

## 4. Discussion

The most suitable habitats for *C. mairei* H. Lév are located in the Yunnan Province, Guizhou Province, Sichuan Province, and Chongqing City, China (southwestern part of China), and to a lesser extent in the Xizang Province and Shaanxi Province, China. The moderately suitable habitat district largely overlaps with the highly suitable habitat district, and a small portion is distributed in Guangxi Province, Hunan Province, Hubei Province, and Henan Province. *C. mairei* H. Lév's distribution in China is not only narrow, but also extremely small, with the highly suitable area only being $39.32 \times 10^4$ km$^2$ in size, with this area only accounting for 4.08% of the land area in China. Additionally, the species has high environmental requirements; therefore, it is now an endangered species and a third-class protected plant in China. According to the model, the best growth conditions of *C. mairei* H. Lév are a min. temperature in the coldest month of $-0.63$~$4.36$ °C, a temperature seasonality of 130.67~642.58, a max. temperature in the warmest month of 28.86~45.97 °C, and precipitation in the coldest quarter of 40.12~101.13 mm. *C. mairei* H. Lév has high temperature requirements and prefers a warm and humid climate. Solar radiation and precipitation will increase in the future climate conditions [30], and the average annual temperature will increase by 2.7–2.9 °C in northern China and by 1.8–2.5 °C in southern China [31]. Appropriate temperature and precipitation are important factors affecting species distribution [32] and are necessary for suitable *C. mairei* H. Lév growth in the future. Temperature seasonality represents the temperature variation over the course of a year, and the greater the standard deviation is, the greater the coefficient of variation is [32]; therefore, this species is not suitable for an environment with a large annual temperature difference.

In the future climate scenario, the suitable area for *C. mairei* H. Lév will increase slightly, and the main reason for this is that temperature and precipitation have a certain influence on the distribution of *C. mairei* H. Lév, and the center of the highly suitable area for *C. mairei* H. Lév growth will change to some extent. In the current climate, the center of the most suitable area is in the Sichuan Province, China, at the geographic coordinates of 105.047° E and 28.768° N. Under the SSP585 climate model, the center of mass will move to the north of China, while in the SSP245 and SSP126 climate models, it will mainly move to the northeast of China. The results show that the distribution of *C. mairei* H. Lév will tend to migrate to higher latitudes under the three climate models.

The MaxEnt model was used according to the distribution characteristics of *C. mairei* H. Lév because the MaxEnt model had a more accurate prediction ability when there were fewer species distribution points [13,33]. There are also some other studies on the prediction of Chinese medicinal plants using the MaxEnt model [34,35], but this model does not take into account human factors, biological factors, or other variables [36] and only takes into account the climate and geographical factors that are the most important for plant growth and distribution. Therefore, considering the development and utilization of medicinal plants by human beings, the actual distribution will decrease to a certain extent.

The conservation and sustainable use of biodiversity is an important task of nature conservation, especially the conservation of rare and endangered plant resources [37]. Studies have proved that the greatest threat to biodiversity is habitat loss, and the most critical means to protect biodiversity is habitat conservation [38]. Based on the results of our study and referring to the research results of other endangered plants [39,40], the conservation of *C. mairei* H. Lév can be carried out in the following aspects. Firstly, in situ conservation by establishing administrative level nature reserves in Chongqing, Yunnan, Sichuan, and Guizhou in China. In medium and poor habitats, decisive conservation measures should be taken because the germplasm resources of *C. mairei* H. Lév are already very scarce. The *C. mairei* H. Lév can be transplanted to areas suitable for its survival that are free from human interference according to suitable environmental conditions such as altitude, temperature, and precipitation, to achieve ex situ conservation of its genetic resources. Finally, the relevant departments should strengthen the publicity and popularize the value of the genetic germplasm resources of *C. mairei* H. Lév, to raise the public awareness of its conservation.

## 5. Conclusions

Under the current climate model, the suitable ecological environment of *C. mairei* H. Lév is a min. temperature in the coldest month of $-0.63\sim4.36$ °C, a temperature seasonality of $130.67\sim642.58$, a max. temperature in the warmest month of $28.86\sim45.97$ °C, and precipitation in the coldest quarter of $40.12\sim101.13$ mm. The most suitable habitats for *C. mairei* H. Lév are mainly located in Yunnan Province, Guizhou Province, Sichuan Province, and Chongqing City, China (southwestern part of China), and to a lesser extent in Xizang Province and Shaanxi Province, China. The moderately suitable habitat district overlaps with the highly suitable habitat, and a small portion of the suitable area is distributed in Guangxi Province, Hunan Province, Hubei Province, and Henan Province. In the future, the size of the highly suitable growing area for *C. mairei* will increase slightly, and the gravity points will shift toward northeast China. It is proposed to establish nature reserves for *C. mairei* H. Lév in the Chongqing, Yunnan, Sichuan and Guizhou provinces of China.

**Author Contributions:** Conceptualization, S.Q. and D.-M.Y.; methodology, S.Q.; software, S.Q. and K.-L.C.; validation, D.-M.Y. and H.-L.L.; formal analysis, W.L. and X.L.; investigation, S.Q.; resources, D.-M.Y.; writing—original draft preparation, W.L. and S.Q.; writing—review and editing, S.Q. and D.-M.Y.; visualization, S.Q.; project administration, D.-M.Y. and Z.-Q.Y.; funding acquisition, D.-M.Y. All authors have read and agreed to the published version of the manuscript.

**Funding:** This work was supported by the National Science Foundation of China (Grant No. 31701963), the Shanghai Agriculture Applied Technology Development Program, China (Grant No. 2021-02-08-00-12-F00756), and the Science and Technology Talent Development Fund for Young and Middle-aged Teachers in Shanghai Institute of Technology (Grant number ZQ2021-21).

**Institutional Review Board Statement:** Not applicable.

**Informed Consent Statement:** Not applicable.

**Data Availability Statement:** Not applicable.

**Acknowledgments:** Thank you to Kaitao Yu for his moral support and encouragement during the writing of this thesis.

**Conflicts of Interest:** The authors declare no conflict of interest.

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
