# Peer review of "The Prediction of the Potentially Suitable Distribution Area of Cinnamomum mairei H. Lév in China Based on the MaxEnt Model"

_sustainability, doi:10.3390/su14137682_

Round 1

Reviewer 1 Report

The manuscript ‘Prediction of the potentially suitable distribution area of Cinnamomum mairei Levl in China based on MaxEnt model’ deals with predicting the habitat suitability of Cinnamomum mairei in China. The manuscript concludes that Our study is of great significance

The study design and used environmental predictors have been selected carefully. The modeling procedure is sound and paper deserves publication in Sustainability. However, following improvements are needed to bring the manuscript in publishable form.

·       The abstract has been concluded with some general messages for all rare medicinal plants. The authors should provide some solid messages regarding the species under question and identify the key hotspots of the species in study region which would be extremely important for species’ conservation in changing climate.

·       I have two major questions with the modeling procedure. The one is what is the extent of global distribution of the species? Why was the model not fitted globally and then downscaled to China? The other is the use of environmental predictors, why geographic and soil factors were excluded. There should be an explanation for this in the MM section.

·       The study seems determining the impact of climate change on the target species. However, the authors have not conveyed their message well. Please explain this in the rationale of the manuscript. Nevertheless, the introduction is too short. It must be extended by 50%.

·       Why authors have not chosen an ensemble modeling approach, although they introduced few SDM algorithms in the introduction section.

·       The studied species, its global habitat and reasons to conserve the species must be stressed in the introduction section

·       Table 1 is unnecessary as all modeling experts have idea what the bios meant for. Please provide correlation matrix of the selected predictors to support the claim that they have less important autocorrelation.

·       The language of the manuscript is too poor. For example section 2.3.

·       Table 2 can also be explained in 1-2 sentences and a reference is needed

·       ROC must be explained in the figure legend of figure 2

·       The unsuitable areas could be changed to white color in all figures

·       /104km2 – the supersripts must be used for better understanding

·       The conclusion must be redrawn based on the values where the desired ecological requirements can be met fort he species

·       The manuscript can be accepted after these changes

Reviewer 2 Report

The topic of the paper is very interesting, but you need to write it better....

Author Response

Thank you for your suggestions, all of them are very important and they will guide me in my future research work.

Point 1:

Referee: The topic of the paper is very interesting, but you need to write it better....

Reply: We carefully revised the full text and submitted it to a professional firm for language enhancement.

Reviewer 3 Report

In the Introduction need to present more information about requires of species (as soil, microclimatic conditions, potential disease etc.) because it can be limiting factors. Diversity of some species can be very limited by the soil conditions and in this research it is unmentioned. Human factors are shyly mentioned. . . Authors need to provide better predictions. Discussion need to be wider and more comprehensive.

Conclusion is completely unclear and unusable in this form. It is one very long sentence. . . Authors need to be very clear about final conclusions.

The manuscript need to be very carefully proofread. In many places I noticed proofreading errors – as missed spaces and points on the end of sentence (lines 34, 36, 43, 45, 48, 56, 117, 118, 124, 171, 172, 230, 231, 248, 249…). Please, carefully check the text, especially the References.

What is “ROC curve”? (line 107)

Climate factors have one mark type in the Tables and another in the text. . . It need be uniformed. (lines 146-148)

There are some sentences which are very confused and obscured and it need be reformulated. (lines 219 – 224).

Author Response

Thank you for your suggestions, all of them are very important and they will guide me in my future research work.

Point 1:

Referee: In the Introduction need to present more information about requires of species (as soil, microclimatic conditions, potential disease etc.) because it can be limiting factors. Diversity of some species can be very limited by the soil conditions and in this research it is unmentioned. Human factors are shyly mentioned. . . Authors need to provide better predictions. Discussion need to be wider and more comprehensive.

Reply: This species has been less studied, and therefore, its suitable environmental factors have not been studied. The focus of our research is to study what are the suitable environmental factors for this species, to understand the suitable environment for the survival of this species through modeling, and to identify the environmental factors that mainly affect its survival. In this study, we did not choose soil and human factors because we need to understand the distribution of the species in the future, and the only data available for the future are climate data. Soil, human factors and other data are not available. We want to explore the trend of the species from the present to the future, so we have selected climate data uniformly.

Based on your comments, we have included this section in the introduction and discussion to give the reader a clearer understanding of the purpose and significance of the study.

Point 2:

Referee: Conclusion is completely unclear and unusable in this form. It is one very long sentence. . . Authors need to be very clear about final conclusions.

Reply:The conclusion was rephrased to make the article more complete.

Point 3:

Referee: The manuscript need to be very carefully proofread. In many places I noticed proofreading errors – as missed spaces and points on the end of sentence (lines 34, 36, 43, 45, 48, 56, 117, 118, 124, 171, 172, 230, 231, 248, 249…). Please, carefully check the text, especially the References.

Reply: We have carefully scrutinized the text and appreciate your comments.

Point 4:

Referee: What is “ROC curve”? (line 107)

Reply: We have added explanations and cited references in the text.

Point 5:

Referee: Climate factors have one mark type in the Tables and another in the text. . . It need be uniformed. (lines 146-148)

Reply: We have unified the writing of climate factors, thank you for your comments.

Point 6:

Referee: There are some sentences which are very confused and obscured and it need be reformulated. (lines 219 – 224).

Reply: We re-wrote the text and sent it to a professional company for language revision to make it more authentic and professional.

Reviewer 4 Report

Dear authors, it was a pleasure to have the opportunity to review the paper Prediction of the potentially suitable distribution area of Cinnamomum mairei Levl in China based on MaxEnt model. The subject is very important, as assessing the habitat suitability and distribution of an endangered species is of high conservation priority. This paper is of great importance locale, but in order to make it more internationally significant, please consider adding reference to the similar studies from all over the World.

Please pay more attention to grammar, sentence construction, and punctuation.

L45-46: please use different sentence construction to emphasize the importance of species conservation and the use of SDMs;

L50: species occurrences;

L51-54: please emphasize the technology behind MaxEnt model (machine learning);

L55-59: it could be useful to international readers to include information from the IUCN red list;

Table 2: for the values 0.7~0.9, maybe more suitable would be "satisfying performance";

L116: please elaborate on which IPCC classification are you referring to;

L121: please make this paragraph more understandable- make a clear distinction between SS pathways, emission scenarios, and other data used, as well as databases and climate models;

L164-166 this part should belong to the Method section;

L181: please be consistent throughout the text, use either Latin or common name of the species, but not both;

L182-183 the same comment is for "China's Silver leaf laurel" (not specified anywhere else in the text);

L219-225: This section is highly un-understandable, please check grammar and sentence construction, maybe divide into several sentences;

L228-229: " Cinnamomum mairei has high requirements on temperature and prefers a warm and humid climate" - maybe it would be more suitable to place this information in the Introduction section (L55-59);

L250: Reference 27 is missing, but regardless- how come the model does not take into account  the human factors and "other variables" when SSPs are based on socio-economic factors. Please make this remark more clear.

Author Response

Thank you for your suggestions, all of them are very important and they will guide me in my future research work.

Point 1:

Referee: Please pay more attention to grammar, sentence construction, and punctuation.

Reply: We performed a full-text revision and submitted it to a professional language company for an upgrade revision.

Point 2:

Referee: L45-46: please use different sentence construction to emphasize the importance of species conservation and the use of SDMs;

Reply: We have modified this section.

Point 3:

Referee: L50: species occurrences;

Reply: We added the global distribution of the species

Point 4:

Referee: L51-54: please emphasize the technology behind MaxEnt model (machine learning);

Reply: We have added the technology behind the MaxEnt model (machine learning) in the preamble.

Point 5:

Referee:  L55-59: it could be useful to international readers to include information from the IUCN red list;

Reply: We have added the IUCN red list information in the introduction.

  1. mairei Levl has most recently been assessed for the international union for conservation of nature (IUCN) Red List of Threatened Species in 1998 (https://www.iucn.org/). The C. mairei Levl is listed as Endangered under criteria B1+2c.)

Point 6:

Referee: Table 2: for the values 0.7~0.9, maybe more suitable would be "satisfying performance"

Reply: We have made changes in accordance with your comments.

Point 7:

Referee: L116: please elaborate on which IPCC classification are you referring to

Reply: We supplemented our test methods with a detailed IPCC taxonomy and cited references.

Point 8:

Referee: L121: please make this paragraph more understandable- make a clear distinction between SS pathways, emission scenarios, and other data used, as well as databases and climate models

Reply: We have added the corresponding notes in the text. The new Intergovernmental Panel on Climate Change (IPCC) has proposed shared socioeconomic pathways (SSPs) to project climate change scenarios. SSPs are designed based on a global development framework. They have been widely used on the global scale to project social and climate scenarios, which got a good result for comprehending future uncertainties. The SSPs include the four emissions scenarios: (1) Radiative forcing stabilizes at 8.5 W m2 in 2100 (SSP5-8.5); (2) Radiative forcing stabilizes at 7.0 W m2 in 2100 (SSP3-7.0); (3) Radiative forcing stabilizes at 4.5 W m2 in 2100 (SSP2-4.5); (4) Radiative forcing stabilizes at 2.6 W m2 in 2100 (SSP1-2.6). Compared with the other future climate scenario analysis tools, SSPs has some advantages: (1) SSPs can comprehensively reflect the complexity of regional development, as their path takes into account various factors of local circumstances, such as population, economy, policy, technology, environment and resources. (2) SSPs in regional impact assessment models usually have high resolution and corrected by historical data. (3) SSPs have better precision than those of global models. Therefore, to determine the future distribution of the species under different climate trajectories, the habitat suitability distribution of C. mairei Levl in the 2030s (2021–2040), 2050s (2041–2060) and 2070s (2061–2080) is simulated with SSP1-2.6, SSP2-4.5 and SSP5- 8.5.

Point 9:

Referee: L164-166 this part should belong to the Method section;

Reply: We put this part into the experimental method.

Point 10:

Referee: L181: please be consistent throughout the text, use either Latin or common name of the species, but not both;

Reply: We have unified the use of either Latin.

Point 11:

Referee: L182-183 the same comment is for "China's Silver leaf laurel" (not specified anywhere else in the text);

Reply: We have unified the use of either Latin.

Point 12:

Referee: L219-225: This section is highly un-understandable, please check grammar and sentence construction, maybe divide into several sentences;

Reply: We reorganized the language and checked the grammar and sentence structure.

Point 13:

Referee: L228-229: " Cinnamomum mairei has high requirements on temperature and prefers a warm and humid climate" - maybe it would be more suitable to place this information in the Introduction section (L55-59);

Reply: No article has been reported on the effect of temperature and humidity on the survival of Cinnamomum mairei. Therefore, the conclusion that "Cinnamomum mairei has high requirements on temperature and prefers a warm and humid climate" is the result of our model validation. We consider adding this part to the introduction.

Point 14:

Referee: L250: Reference 27 is missing, but regardless- how come the model does not take into account the human factors and "other variables" when SSPs are b ased on socio-economic factors. Please make this remark more clear.

Reply: We have added references. Our main focus of this study is on the impact of climate on the species and predicting future trends in the habitat of the species as the climate changes. For the time being, we only have access to climate data, but not to future demographic, economic, policy, technological, environmental, and resource data. This is worthy of study by later scholars.

Round 2

Reviewer 1 Report

Cinnamomum mairei H.Lév. is an accepted name of the species. Please correct it throughout the manuscript.

Please use author name with species at first use only.

The abstract says 21 geographic variables were used to predict the species distribution, which is incorrect.

The conclusion is again not convincing. There should be guidelines that where the species would be safe under changing climate conditions

It is better to write total suitable area as well in the provinces predicted suitable for the species

The language is still poor, although authors claim that the manuscript was edited by an editing service

ROC curve of 10 simulations. – Again ROC has not been explained in the figure legend

The discussion section is short and needs considerable extension.

In short the desired corrections were not satisfactory

Author Response

Thank you for your suggestions, all of them are very important and they will guide me in my future research work.

Point 1:

Referee: Cinnamomum mairei H. Lév. is an accepted name of the species. Please correct it throughout the manuscript.

Reply: Thank you for your comments, we have made changes to the full text.

Point 2:

Referee: Please use author name with species at first use only.

Reply: We have made changes

Point 3:

Referee: The abstract says 21 geographic variables were used to predict the species distribution, which is incorrect.

Reply: We changed it to 6 environmental factors (temperature seasonality, max temperature in the warmest month, min temperature in the coldest Month, precipitation seasonality, precipitation in the coldest quarter and aspect)

Point 4:

Referee: The conclusion is again not convincing. There should be guidelines that where the species would be safe under changing climate conditions.

Reply: We have improved the content of this section.

Point 5:

Referee: It is better to write total suitable area as well in the provinces predicted suitable for the species.

Reply: We mainly want to understand the change trend and distribution pattern of species under climate change, so we did not add the area of fitness zones in each province, but have the change of area of each fitness zone in China. We refer to several excellent papers, which are roughly similar to mine.

Refer:

    1. Xing Yu, Xu Tao, et al. Predicting potential cultivation region and paddy area for ratoon rice production in China using Maxent model, Field Crops Research, Volume 275, 2022, 108372. https://doi.org/10.1016/j.fcr.2021.108372.

  1. Peng Zhan, Feiyang Wang, et al. Assessment of suitable cultivation region for Panax notoginseng under different climatic conditions using MaxEnt model and high-performance liquid chromatography in China, Industrial Crops and Products, Volume 176, 2022, 114416. https://doi.org/10.1016/j.indcrop.2021.114416.

Point 6:

Referee: The language is still poor, although authors claim that the manuscript was edited by an editing service

Reply: Our articles have been given to MDPI journals for language touch-ups, and we regret if they do not meet your expectations.

Point 7:

Referee: ROC curve of 10 simulations. – Again ROC has not been explained in the figure legend.

Reply: We have added an explanation in the figure legend.

Point 8:

Referee: The discussion section is short and needs considerable extension.

Reply: We have added content to the discussion section

Reviewer 2 Report

The paper is much improved.

Author Response

Thanks to the reviewers' comments, the article has been greatly improved.

Reviewer 4 Report

Dear authors,

thank you for improving your manuscript significantly.

You responded very well and argumented to all of the remarks.

The only thing that caught my eye is 

L298: instead of Chinese herbal medicine please consider using Chinese medicinal plants or similar. Also please check the reference 36, if I'm not mistaken it refers to the use of MaxEnt in India.

Author Response

Dear reviewer, your comments were excellent and we replaced the words.

The object of study in reference 36, Perilla frutescens, is an Chinese medicinal plant, and although its study area is India, the text mainly wants to express the study of maxent model on Chinese medicinal plants prediction.